# The Molecular Characterization of a New *Prunus*-Infecting Cheravirus and Complete Genome Sequence of Stocky Prune Virus

**DOI:** 10.3390/v14112325

**Published:** 2022-10-23

**Authors:** Maryam Khalili, Thierry Candresse, Yoann Brans, Chantal Faure, Jean-Marc Audergon, Véronique Decroocq, Guillaume Roch, Armelle Marais

**Affiliations:** 1Univ. Bordeaux, INRAE, UMR BFP, 33140 Villenave d’Ornon, France; 2Laboratoire de Virologie et de Biologie Moléculaire, CTIFL, 24130 Prigonrieux, France; 3UR GAFL, INRAE, 84143 Montfavet Cedex, France

**Keywords:** high-throughput sequencing, stone fruit, phylogenetic analysis, prunus, cheravirus, stocky prune virus, alpine wild prunus virus

## Abstract

As part of a virome characterization of *Prunus* species, a novel cheravirus was discovered in two wild species, *Prunus brigantina* and *P. mahaleb*, and in an apricot (*P. armeniaca*) accession. The sequence of the two genomic RNAs was completed for two isolates. The Pro-Pol conserved region showed 86% amino acid (aa) identity with the corresponding region of trillium govanianum cheravirus (TgCV), a tentative *Cheravirus* member, whereas the combined coat proteins (CPs) shared only 40% aa identity with TgCV CPs, well below the species demarcation threshold for the genus. This suggests that the new virus should be considered a new species for which the name alpine wild prunus virus (AWPV) is proposed. In parallel, the complete genome sequence of stocky prune virus (StPV), a poorly known cheravirus for which only partial sequences were available, was determined. A phylogenetic analysis showed that AWPV, TgCV and StPV form a distinct cluster, away from other cheraviruses.

## 1. Introduction

The genus *Cheravirus* is one of the nine genera (according to the last update of https://talk.ictvonline.org/taxonomy/ verified on 3 August 2022) assigned to the family *Secoviridae*. The recognized species include *Apple latent spherical virus* (ALSV), *Cherry rasp leaf virus* (CRLV), *Arracacha virus B* (AVB), *Currant latent virus* (CuLV) and *Stocky prune virus* (StPV), while two tentative members have recently been discovered, trillium govanianum cheravirus (TgCV) and babaco cheravirus 1 (BabChV-1) [1,2,3,4,5,6,7,8,9]. Cheraviruses have bipartite, single-stranded positive-sense RNA genomes. Full-length recognized cheravirus RNA1 molecules have been determined in a range of 6.8–7.1 kb and RNA2 molecules in a range of 3.2–3.7 kb-long. Each genomic RNA encodes a polyprotein (RNA1: P1; RNA2: P2) that is cleaved by the RNA1-encoded 3C proteinase to generate the functional non-structural and structural proteins. Members of the genus *Cheravirus* encode three capsid proteins. Some are known to be transmitted by nematodes (CRLV) and aphids (CuLV). Seed transmissibility was observed in CRLV, ALSV and AVB, while ALSV and AVB are also pollen-transmissible [8,10,11,12,13].

Stone fruits are deciduous trees native to the temperate zone of the northern hemisphere. All stone fruits such as apricot, plum, peach/nectarine and sweet and sour cherry alongside almond belong to the genus *Prunus* of the *Rosaceae* family and can be affected by various viral diseases. CRLV and StPV are the two known *Cheravirus* members naturally infecting *Prunus*. Sweet cherry (*Prunus avium*), peach (*P. persica*) and susceptible cherry rootstocks including both widely used “Mazzard” (*P. avium*) and “Mahaleb” (*P*. *mahaleb*) are the main *Prunus* hosts for CRLV [14,15]. StPV is the causal agent of the stocky prune disease that affects European plum (*P. domestica*) and that is characterized by severe symptoms of shortened internodes, chlorotic, rolled and enlarged leaves, premature fruit fall and tree decline over time [3,4,16]. The disease, geographically limited to the southwest of France, resulted in a notable loss of production. It was a source of concern at the beginning of the 20th century, but its impact has fortunately massively declined, probably as a consequence of the sanitary selection of planting materials. To date, only partial sequence data representing the 3′ part of RNA1 (NC_043387, 2644 nt) and RNA2 (NC_043388, 1794 nt) are available [4].

As part of a virus discovery effort in *Prunus* crops, the virome of over 300 samples belonging to 9 crop or ornamental species *(P. domestica*, *P. avium*, *P. cerasus*, *P. armeniaca*, *P. persica*, *P. dulcis*, *P. serrulata*, *P. nipponica*, and *P. subcordata*), as well as over 300 samples of wild *Prunus* species (*P. brigantina*, *P*. *mahaleb*, *P. spinosa*, *P. serotina* and *P. cerasifera*) were investigated using high-throughput sequencing (HTS). In this study, we present the molecular characterization of a new *Prunus*-infecting cheravirus identified in two wild *Prunus* species and in an apricot accession and the determination of the complete genome sequence of StPV.

## 2. Materials and Methods

### 2.1. Plant Material

Pooled leaf samples collected from different parts of the canopy, regardless of symptoms, were collected from 50 apricot (*P. armeniaca*) varieties originated from 16 countries kept in collection at INRAE *Prunus* Biological Resource Center (BRC, INRAE, Avignon, France). In particular, the apricot accession A1915 found to be infected by a novel cheravirus is an old variety (San Castrese) from the Campania region in southern Italy, introduced from Venturina Pisa University experimental repository in the collection. In addition, similar leaf samples were collected in the French Alps in 2017 [17] and 2021, respectively, for wild growing *P. brigantina* and *P. mahaleb* trees. The original, partially sequenced StPV isolate [4] was maintained in GF305 peach seedlings in collection at INRAE virus repository under greenhouse conditions. In all cases, fresh leaf tissues were stored at −80 °C or dried over anhydrous CaCl_2_ (Sigma Aldrich Chimie, Saint-Quentin-Fallavier, France) and preserved at room temperature until use.

### 2.2. Double-Stranded RNA Extraction, Library Preparation and High-Throughput Sequencing (HTS)

Double-stranded RNA were purified from dried leaves of *P. brigantina* or of the 50 apricot accessions following the method of [18] before being used for cDNA synthesis using SuperscriptII Reverse Transcriptase and LDF primers [19] according to manufacturer’s instructions (Invitrogen/Fisher Scientific, Illkirch, France). A random PCR amplification was performed on each cDNA preparation using multiplex identifier (MID) adaptors and Dream Taq DNA polymerase (Thermo Fisher Scientific) as described by [18], enabling all samples to be sequenced in a multiplexed format. PCR products purified using a MiniElute PCR Purification Kit (Qiagen SAS France, Courtaboeuf, France) were pooled in equimolar amounts prior to library preparation using the TruSeq Nano kit (Illumina) and Illumina sequencing on a Hiseq3000 platform (2 × 150 bp) [outsourced at the GetPlage INRAE platform (Toulouse, France) or Azenta (Leipzig, Germany)].

### 2.3. Total RNA (totRNA) Extraction and HTS

Total RNA HTS RNAseq was performed for a *P. mahaleb* sample and the StPV-inoculated GF305 sample. Total RNA was extracted from fresh leaves using a modified CTAB procedure [20] and sequenced after a ribodepletion step in a HiSeq3000 2 × 150 bp format (Azenta).

### 2.4. Total Nucleic Acids (TNAs) Extraction and RT-PCR Detection of the Novel Cheravirus

TNAs were extracted from graft-inoculated GF305 leaves as well as leaves from noninoculated controls three months after the grafting assay (see below) using protocol 1 described in [21]. Specific primers allowing the detection of AWPV-Pa were designed from contigs in RNA1 (Seco-PA-R1-F/R, Appendix A) and used to perform a two-step RT-PCR assay as already described [22].

### 2.5. HTS Data Analyses

Sequencing reads were quality controlled and trimmed before de novo assembly using CLC Genomic Workbench version 21.0.3 (Qiagen). A BlastX analysis to identify viral contigs was performed against the GenBank protein database (nr) limited to viruses. The contigs detected as representing new viral sequences were extended by rounds of mapping residual reads. If needed, extended contigs were manually scaffolded before closing any residual small gaps by RT-PCR and the Sanger sequencing of amplicons. An additional analysis was conducted by mapping HTS reads on a set of *Prunus* viruses reference genomes (min length fraction = 0.8; min similarity fraction = 0.7).

### 2.6. Genome Sequence Completion for Isolates of the Novel Cheravirus and StPV

Based on the scaffold sequence reconstructed for the RNA1 of the novel cheravirus *P. mahaleb* isolate, a primer pair (Appendix A) was designed to amplify a cDNA fragment spanning the remaining small gap in the sequence assembled for the *P. brigantina* isolate, using a two-step RT-PCR assay as described in [22]. The 5′ ends of the two genomic RNAs of StPV and of the *P. brigantina* and *P. mahaleb* isolates of the novel cheravirus were obtained by the rapid amplification of cDNA ends (RACE) experiments using the SMARTer^®^ RACE 5′/3′Kit (Takara Bio Europe SAS, Saint-Germain-en-Laye, France) on either dsRNA or totRNA templates using primers designed based on the assembled HTS sequences (Appendix A) and following the manufacturer’s instructions. Additionally, the 3′ ends were amplified using contig-based designed forward primers together with LD-prim primers (Appendix A) through long-distance (LD) RT-PCR according to the protocol already described [23]. Amplicons were directly Sanger sequenced on both strands (Eurofins Genomics, Ebersberg, Germany).

### 2.7. Pairwise Comparisons, Phylogenetic and Recombination Analyses

Multiple alignments of nucleotide (nt) or amino acid (aa) sequences were performed using the ClustalW program [24] in MEGA 7 [25]. The aa alignments were used for maximum likelihood phylogenetic analyses using the RtRev +I+G evolutionary model. Branch validity was evaluated by randomized bootstrapping (1000 replicates). Conserved protein domains in the aa deduced sequences were searched using the CD search tool (https://www.ncbi.nlm.nih.gov/Structure/cdd/wrpsb.cgi, accessed on 22 August 2022). Recombination analysis was performed using alignments of RNA1 or RNA2 of cheraviruses and the RDP4 program with default parameters [26].

### 2.8. Biological Indexing in Greenhouse

Wood samples from the apricot accession A1915 were sampled to conduct a biological indexing in greenhouse. Part of the tree showed leaf deformation and chlorosis symptoms. Biological indexing on peach GF305 with chip-budding graft inoculation was conducted [27]. Two weeks after grafting, the GF305 indicators were cut off, and ten replicates (five from symptomatic and five from non symptomatic branches) were monitored over several weeks to spot the appearance of any virus-like symptoms on the growth and leaves of the indicator plants in comparison to negative controls (non inoculated GF305).

## 3. Results

### 3.1. Determination of the Complete Genome Sequence of a Newly Discovered Cheravirus and of StPV

Of several *P. brigantina* samples analyzed by dsRNA-based HTS, one revealed the presence of a novel *Secoviridae*. After demultiplexing, quality trimming and the de novo assembly of the 18,93,230 reads obtained for this sample, BlastX-based contigs annotation allowed us to identify contigs with significant identity with members of the *Secoviridae* family. Scaffolds for both genomic RNAs were then reconstructed using the contigs of interest. Consecutive rounds of reads mapping to the scaffolds extended the sequences to yield near-complete viral molecules. A small gap in RNA1 scaffold was filled by the direct sequencing of a RT-PCR fragment generated from dsRNA using a primer pair flanking the gap (Appendix A). Overall, the reads assembled in the completed sequences represented 30.5% and 22% of total reads for RNA1 and RNA2, respectively, corresponding to average coverages of 8,886× (RNA1) and 15,047× (RNA2) (Table 1). In Blast searches of the GenBank database, the most closely related virus was trillium govanianum cheravirus (TgCV), a tentative member in the *Cheravirus* genus with 54% nt identity in RNA1 and 56.2% nt identity in RNA2. When used as references, the scaffolds assembled for the *P. brigantina* virus enabled us to identify two other related viruses in two HTS datasets: one from a *P. mahaleb* tree from the French Alps, one for which two long contigs representing nearly complete genomic sequences (RNA1 and 2) were obtained at high coverage (Table 1), and the other from an apricot (*P. armeniaca*) sample from Prunus BRC for which the assembly was much less complete. For the RNA1 of the isolate in the apricot sample, 19 small nonoverlapping contigs were obtained, covering an overall 4650 nt of the RNA1 molecule. The RNA2 molecule was even more fragmented. Further analyses revealed that the two isolates identified in *P.*
*brigantina* and *P. mahaleb* share 86.3% nt identity in their RNA1 (93.7% aa identity in the encoded polyprotein) and 86.7% nt identity in their RNA2 (93.6% aa identity in the polyprotein). More precisely, in the two regions used for molecular species demarcation in the *Secoviridae* family (Pro-Pol region and combined CPs), the levels of identity between the two isolates are well above the cut-off (less than 80% or 75% aa identity, respectively, [28]), with respectively 97.1% and 93.9% aa identity (Table 2). In parallel, similar pairwise comparisons performed with the assembled scaffold of the RNA1 of the isolate identified in apricot comprising 4650 nt showed that it shares 84.3–85% nt identity with the corresponding region of the RNA1 of the *P. mahaleb* and *P. brigantina* isolates (93.6–93.2% for the encoded polyprotein fragment). These findings show that the three identified viruses are isolates of a novel *Cheravirus* species that naturally infects different *Prunus* hosts and tentatively named alpine wild prunus virus (AWPV). RACE experiments were performed to determine the 5′ and 3′ end sequences for both genomic RNAs of the *P. mahaleb* and *P. brigantina* isolates, while no specific effort was made to complete the genome sequence of the apricot isolate. The full-length (excluding polyA tail) RNA1 of AWPV in *P. brigantina* and *P. mahaleb* was determined to be 7354 nt and 7491 nt, respectively and the RNA2 was 3574 nt and 3568 nt long, respectively (Table 3). The complete genome sequences have been deposited in GenBank under the accession numbers OP328247-8 for AWPV from *P. brigantina* (AWPV-Pb), and OP328249-50 for AWPV from *P. mahaleb* (AWPV-Pm) (Table 1).

In parallel, a total of 57,041,248 paired-end Illumina reads were recovered for the StPV graft-inoculated GF305 sample (Table 1). Following the de novo assembly of trimmed reads and BlastX analyses, two long contigs of nearly complete genome length were identified for the genomic RNAs of StPV, with over 4000x coverage (Table 1). Both sequences showed 99.9% nt identity with the previously determined partial StPV RNA1 and RNA2 sequences (NC_043387-88). The 5’ and 3’ ends were determined by RACE or polyA-anchored PCR for each molecule. The complete StPV RNA1 (7021 nt-long) and RNA2 (3495 nt-long) sequences have been deposited in GenBank under accession numbers OP328251 and OP328252, respectively (Table 1).

### 3.2. Genomic Organization of AWPV and StPV and Determination of Their Phylogenetic Relationships

Based on the known genomic organization of other cheraviruses and on sequence homologies with them, a genomic organization of AWPV and StPV was proposed (Figure 1A,B) in which polyprotein 1 (P1) is cleaved in six mature proteins, while polyprotein 2 (P2) is cleaved into four proteins including a movement protein and three coat proteins. However, the analysis of polyprotein alignments and the variety of cleavage sites already reported for cheraviruses [1,2,6,7,8] did not allow for predicting cleavage sites for StPV and AWPV polyproteins. A single open reading frame (ORF) is identified in both RNA1 and RNA2 (Figure 1A,B). The AUG start codon at nt positions 51–53 in RNA1 AWPV-Pb and AWPV-Pm (nt 81–83 in RNA1 StPV) is in favorable translation initiation context, with a purine (A/G) at position -3, [29] for both AWPV and StPV. In the case of RNA2, although the AUG codon in nt positions 70–72, 71–73 and 79–81, in AWPV-Pb, AWPV-Pm and StPV, respectively, is in a suboptimal context for initiating translation, it is considered to be the most likely site of initiation, since 5′ NCRs would then being the same length range in both RNA1 and RNA2, while the use of the next downstream AUG codon (286–288, 287–289 and 255–257, respectively in AWPV-Pb, AWPV-Pm and StPV) would result in 5’ NCRs widely differing in length between the two genomic RNAs. The RNA1 and RNA2 segments of AWPV-Pb and AWPV-Pm are very largely colinear, but in the RNA1 sequence, an indel of 138 nt is located between nt 278 and 416 (referring to the AWPV-Pm RNA1), leading to a deletion of 46 aa in the N-terminal part of deduced P1 polyprotein in AWPV-Pb. The RNA2 also presents a short indel of six nt in the *P. mahaleb* isolate (from nt 2077 to 2083, referring to AWPV-Pb RNA2), leading to a deletion of two aa in the deduced P2 polyprotein in the CPs coding region.

The RNA1 5′ noncoding regions (NCRs) are 50 nt long in AWPV-Pb and AWPV-Pm, and 80 nt long in StPV (Table 3). The 3′ NCRs are longer, comprising between 229 nt (AWPV-Pm) and 251 nt (StPV) (Table 3). In RNA2, the 5′ NCRs are 69, 70 and 78 nt long, and the 3′ NCRs comprise between 241 nt (AWPV-Pb) and 251 nt (AWPV-Pm) (Table 3). As reported for other *Secoviridae* members with bipartite genomes, sequence homologies were identified between the NCRs of the two genomic RNAs. The 5′ NCRs show 22 to 28 fully conserved 5’ nucleotides, depending on the virus considered, whereas the 3′ NCRs display 89.5%, 97.8% and 98.9% sequence identity between the two genomic RNAs of StPV, AWPV-Pb, and AWPV-Pm, respectively.

As expected, two conserved domains were identified within the P1 polyprotein of both AWPV and StPV. The first was an RNA helicase domain (pfam 00910) located between aa 849 and 949 in AWPV-Pb. Corresponding values for AWPV-Pm and StPV are, respectively, 895–995 and 835–933. The second was an RNA-dependent RNA polymerase domain (cd01699) between aa 1731–2027 for AWPV-Pb and aa 1777–2073 or 1712–2011 for AWPV-Pm and StPV, respectively. No conserved domains could be identified in the P2 polyproteins, which is in keeping with the very low percentages of identity observed between the CPs of different cheraviruses (Table 2).

An analysis of potential recombination events in the genomes of *Cheravirus* members or potential members using RDP4 [26] failed to identify any significant events. In order to confirm the taxonomical position of AWPV, phylogenetic analyses based on the P1 and P2 polyproteins of representative *Cheravirus* members were performed (Figure 2). The phylogenetic trees clearly show the clustering of the two isolates of AWPV together with StPV and TgCV, within a subgroup distinct from the rest of the cheraviruses (Figure 2). The same clustering was obtained when a phylogenetic tree integrating representative members of the various genera in the *Secoviridae* family was generated based on the Pro-Pol region (Appendix A). Pairwise sequence comparisons showed AWPV-Pb Pro-Pol aa sequence shares between 38.7% (with ALSV) and 86.3% (with TgCV) aa identity (Table 2). Separately, the level of identity between CPs is lower, comprising between 12.2% (with AVB) and 41.1% (with TgCV) (Table 2). The pairwise comparisons performed with the AWPV-Pm provided similar values of identity. Concerning StPV, the most closely related cheraviruses are AWPV (Pb and Pm) and TgCV with 64–64.7% and 61.7% aa identity in the Pro-Pol region, respectively, and 33.8–34.1% and 40% aa identity in the CPs region, respectively.

### 3.3. Biological Indexing of AWPV-Infected Apricot Accession

About three months after biological indexing in greenhouse, some GF305 indicator plants showed symptoms of leaf chlorosis, strangulation and deformation. The presence of AWPV-Pa was verified by RT-PCR. All the GF305 plants, with and without symptoms, were found to be infected by AWPV-Pa. The observed symptoms may be associated with the presence of prunus necrotic ringspot virus (PNRSV), which was confirmed by DAS-ELISA testing. These symptoms could therefore not be linked to the presence of AWPV.

## 4. Discussion

Metagenomics have greatly expanded our knowledge of plant viromes by contributing to the discovery of novel viruses. This includes the viromes shared in both wild and cultivated crops. Taking advantage of HTS, we investigated a large number of wild and cultivated *Prunus* samples, leading to the discovery of a putative new cheravirus tentatively named alpine wild prunus virus (AWPV), in three different host species, *P. brigantina* (wild relative of apricot), *P. armeniaca* (cultivated apricot) and *P. mahaleb* (mahaleb cherry or St. Lucie cherry). Furthermore, this paper reports the complete genome sequence of two isolates of AWPV and the completion of the genome sequence of the previously partially characterized StPV [4]. These findings add AWPV to the list of tentative cheraviruses, which already increased over the last few years with CuLV [8], BabChV-1 [5] and TgCV [9].

The size of the AWPV RNA1 is slightly above the expected length (6.8–7.1 kb) of RNA1 in the genus *Cheravirus*. This can be explained by considering that only four complete genomes of cheraviruses had been available when establishing the genome content [28]. Although the percentage of the aa identity between AWPV and TgCV in the Pro-Pol region is above the species demarcation threshold currently accepted in the *Secoviridae* family (86.3% vs. 80%), the level of aa identity in the CPs region is well below the cut-off (41.1% vs. 75%), so that there is no ambiguity that AWPV represents a distinct species.

AWPV was initially identified in a *P. brigantina* tree sampled in 2017 in a rather isolated area of the French Alps. Efforts to resample that tree in 2021 were not successful since the tree had died in the interim period from unknown causes. Sampling in 2021 of *P. mahaleb* trees in a nearby valley, about 25 km away on a direct line, allowed us to identify one tree out of the 10 sampled which was infected by AWPV. AWPV was also later identified in a single apricot tree of the old Italian variety San Castrese. Even though the virus was detected in single infection in *P. brigantina* and in co-infection with another virus (prunus mahaleb-associated luteovirus, PmaLV) in *P. mahaleb* [30], the association of the symptoms with AWPV infection was not possible because of the unusual harsh, high mountain, growth conditions of the trees. Both the dead *P. brigantina* tree and the *P. mahaleb* tree showed a bushy growth and shortened internodes, but it is not possible to know whether these symptoms were caused by AWPV infection or by mountain growth conditions, or in the case of the *P. mahaleb* tree, by the co-infection with PmaLV, a very recent discovered luteovirus [30]. It should be noted, though, that of the 10 *P. mahaleb* trees sampled in 2021, which all showed similar poor, stunted growth, a single one was found infected by AWPV. The leaf deformation and chlorosis observed on the leaves of the AWPV infected apricot could likewise be correlated with its co-infection by PNRSV. The results from the bioassays conducted on samples collected from this apricot tree came to the same conclusion. Further studies would be necessary, especially from AWPV single-infected plant material, to assess the symptomatology associated with this virus. Overall, our observation cannot be used to establish a clear link between AWPV infection and symptoms in *Prunus* hosts.

Unlike AWPV, StPV is known to be the causal agent of a severe disease developing leaf chlorosis and deformation, bushy growth and early drop of the fruits, symptoms diversely called “maladie de Brugères”, “maladie du prunier stérile”, “maladie du prunier mâle” or stocky prune disease [3,4,16]. It also induces particularly severe stunting in the GF305 peach indicator, and there is overall no ambiguity about its pathogenicity in *Prunus*. The completion of its genomic sequence reported here unambiguously confirms that it belongs to the *Cheravirus* genus and demonstrates that it is related to AWPV.

The availability of full genomic sequences facilitates the development of specific and inclusive detection tests, which would permit further studies on the geographical distribution of AWPV, its genetic diversity and its host range. The fact that it was identified in two different *Prunus* species sampled each time as few individuals suggests that it might not be rare in wild *Prunus* hosts in that particular area of the French Alps and possibly elsewhere. This virus was detected in an area with few or no *Prunus* crops, which may have prevented its transfer from wild to cultivated *Prunus* species. Ultimately, this could be one of the reasons it was detected only once among more than 300 cultivated *Prunus* accessions tested by HTS during this survey. In a similar fashion, StPV was not detected among the more than 600 *Prunus* samples analyzed by HTS, which is in keeping with its initial limited distribution and (near) complete disappearance in recent times. Although soil transmission was considered a possibility for StPV, no vector has been identified to date and the transmission mechanism(s) of AWPV remain to be analyzed.

Once again, this study demonstrates the power of HTS approaches for virus discovery. A precise evaluation of the risks potentially associated with AWPV for *Prunus* crops remains to be performed, in particular when it comes to pathogenicity to a range of cultivated *Prunus* species and to the existence and efficiency of potential transmission mechanisms.

## Figures and Tables

**Figure 1 viruses-14-02325-f001:**
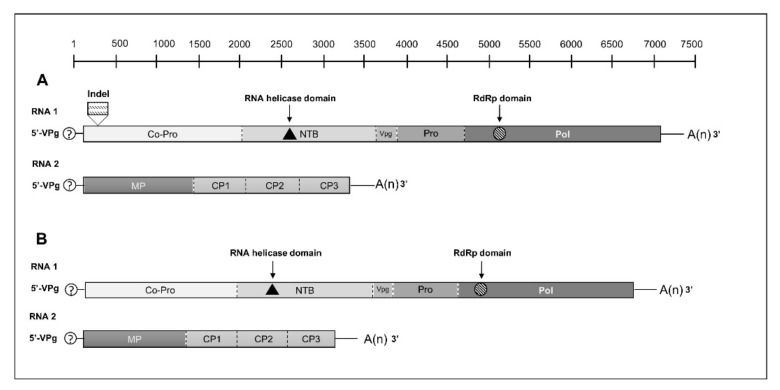
Hypothetical genome organization of alpine wild prunus virus-*P. brigantina* (AWPV-Pb) (**A**) and of stocky prune virus (StPV) (**B**). Putative cleavage sites of RNA1 and 2 deduced polyproteins are represented by dashed lines. The conserved motifs of helicase and RNA-dependent RNA polymerase (RdRp) are shown by black triangle and hatched circle, respectively. The position of the indel of 138 nt in AWPV-Pb RNA1 in comparison to AWPV-Pm RNA1 is shown. Co-Pro: protease cofactor; NTB: nucleotide triphosphate binding helicase; VPg: viral genome-linked protein; Pro: protease; Pol: polymerase; MP: movement protein; CP: coat protein. RdRp: RNA-dependent RNA polymerase.

**Figure 2 viruses-14-02325-f002:**
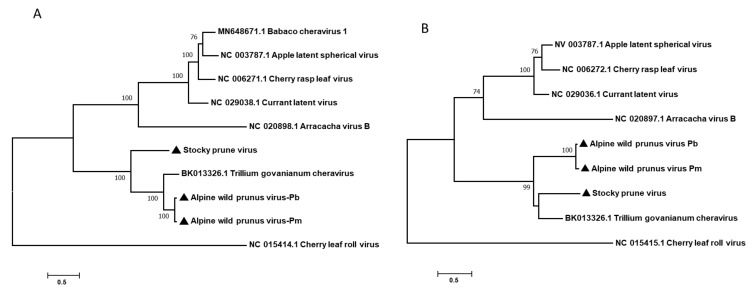
Phylogenetic trees reconstructed using the alignment of amino acid sequences of RNA1-encoded polyprotein P1 (**A**), and RNA2-encoded polyprotein P2 (**B**) of representative members of the genus *Cheravirus*. Trees were generated using the Maximum Likelihood method. The statistical significance of branches was evaluated by bootstrap analysis (1000 replicates). Cherry leaf roll virus (CLRV) was used as an outgroup. Bootstrap values less than 70% were removed. Viruses characterized in the present study are marked by black triangles. AWPV-Pb: alpine wild prunus virus-*P. brigantina*; AWPV-Pm: alpine wild prunus virus prunus-*P. mahaleb*; StPV: stocky prune virus; AVB: arracacha virus B; ALSV: apple latent spherical virus; BabChV-1: babaco cheravirus-1; CRLV: cherry rasp leaf virus; CuLV: currant latent virus; TgCV: trillium govanianum cheravirus. Accession numbers are given before virus abbreviations. The bars represent the genetic distances.

**Table 1 viruses-14-02325-t001:** Target nucleic acids population, number of trimmed reads, average coverage and percentage of total reads mapping to the genomic RNAs of alpine wild prunus virus (AWPV) and stocky prune virus (StPV).

Virus-Isolate ^1^	Method	Trimmed Reads	Segment	Average Coverage	Mapped Reads (%)	Accession Numbers
AWPV-Pb	dsRNA ^2^	1,893,230	RNA1	8886x	30.5%	OP328247
			RNA2	15047x	22%	OP328248
AWPV-Pm	RNA	61,886,192	RNA1	3503x	0.2%	OP328249
			RNA2	10145x	0.39%	OP328250
StPV	RNA	57,041,248	RNA1	4051x	0.34%	OP328251
			RNA2	4476x	0.18%	OP328252

^1^ AWPV-Pb: alpine wild prunus virus-*P. brigantina*; AWPV-Pm: alpine wild prunus virus-*P. mahaleb*; StPV: stocky prune virus. ^2^ dsRNA: double-stranded RNA.

**Table 2 viruses-14-02325-t002:** Percentages of identity observed between the Pro-Pol region and the capsid protein region of alpine wild prunus virus-*P. brigantina* (AWPV-Pb) and the corresponding proteins of *Cheravirus* accepted or tentative members.

Virus ^1^	Amino Acid Identity (%)	
	Pro-Pol ^2^	Capsid Protein ^3^
AWPV-Pm	97.1%	93.9%
StPV	64.7%	33.7%
ALSV	38.7%	16.3%
AVB	41.6%	12.2%
CRLV	38.9%	15.9%
CuLV	39.4%	14.6%
TgCV	86.3%	41.1%
BabChV-1	39.2%	na ^4^

^1^ AVB: arracacha virus B; ALSV: apple latent spherical virus; BabChV-1: babaco cheravirus-1; CRLV: cherry rasp leaf virus; CuLV: currant latent virus; TgCV: trillium govanianum cheravirus. ^2^ The Pro-Pol region is delineated by the “CG” motif of the 3C-like proteinase and the “GDD” motif of the polymerase [28]. ^3^ For viruses with two or three CPs, combined CP sequences are considered [28]. ^4^ not applicable (No RNA2 data available in GenBank).

**Table 3 viruses-14-02325-t003:** Characteristics of the genomic RNAs of the cheraviruses characterized in this study.

Virus	Segment	Length (nt)	Polyprotein (aa)	5′ NCR ^1^ (nt)	3′ NCR (nt)
AWPV-Pb	RNA1	7354	2358	50	230
	RNA2	3574	1088	69	240
AWPV-Pm	RNA1	7491	2404	50	229
	RNA2	3568	1086	70	241
StPV	RNA1	7021	2230	80	251
	RNA2	3495	1056	78	249

^1^ NCR: noncoding region.

## Data Availability

The sequences reported in the present manuscript have been deposited in the GenBank database under the accession numbers OP328247-OP328252.

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
