# Peer review of "The Molecular Characterization of a New Prunus-Infecting Cheravirus and Complete Genome Sequence of Stocky Prune Virus"

_viruses, 2022, doi:10.3390/v14112325_

Round 1

Reviewer 1 Report

The paper by Khalili et al. identified a new species of Cheravirus AWPV from Prunus brigantine, P. mahaleb and P. armeniaca using high-throughput sequencing methods, characterizing the genome sequence and biological property by grafting. However, the main concerns were as below. 

1). Why did the author use different HTS methods (double-stranded RNA and total RNA) for virus detection from P. brigantine and P. mahaleb? 

2). AWPV was discovered in three Prunus species. However, only two complete genomes were determined from P. brigantine and P. mahaleb, while not from P. armeniaca. Why?

3). Why did the author sequence a complete genome of StPV? What’s the relationship of this work with the general objective of the whole article?

Author Response

1). Why did the author use different HTS methods (double-stranded RNA and total RNA) for virus detection from P. brigantine and P. mahaleb? 

Leaves of P. mahaleb sample were collected in 2021 and stored at -80°C, which allowed us to extract total RNA of high quality, as required for RNAseq. In contrast, P. brigantina samples were collected in 2017 (Liu et al., 2021) and leaves had been stored dried over anhydrous CaCl2. Usually, total RNA extraction from dried leaves results in low quality or low yield, but double-stranded RNAs are efficiently purified from such dried materials following the protocol we published (Marais et al., 2018).

Liu, S.; Decroocq, S.; Harte, E.; Tricon, D.; Chague, A.; Balakisiyeva, G.; Kostriesyna, T.; Turdiev, T.; Fisher-Le Saux, M.; Dallot, S.; Girauc, T.; Decrooccq V. Genetic diversity and population structure analyses in the Alpine plum (Prunus brigantina Vill.) confirm its affiliation to the Armeniaca section. TGG 2021, 17, 2. doi: 10.1007/s11295-020-01484-617

Marais, A.; Faure, C.; Bergey, B.; Candresse, T. Viral Double-Stranded RNAs (dsRNAs) from Plants: Alternative Nucleic Acid 437 Substrates for High-Throughput Sequencing. Methods Mol Biol. 2018, 1746, 45-53. doi: 10.1007/978-1-4939-7683-6_4

 2). AWPV was discovered in three Prunus species. However, only two complete genomes were determined from P. brigantine and P. mahaleb, while not from P. armeniaca. Why?

The reads assembly obtained for the P. armeniaca isolate was much less complete, in comparison to the two other isolates, possibly as a consequence of lower virus titer of of the mixed infection status. As explained in the text, 19 small non-overlapping contigs were obtained for the RNA1, covering an overall 4,650 nt of the genomic molecule, whereas the RNA2 molecule was even more fragmented. Given that two complete genomes were already available with the AWPV P. brigantina and P. mahaleb isolates, and that the partial sequence from apricot allowed to reach a conclusion on the relationships between isolates, we felt that generating a full genome for the apricot virus was not so important.

3). Why did the author sequence a complete genome of StPV? What’s the relationship of this work with the general objective of the whole article?

The stocky prune virus is a very poor known Cheravirus, for which only a partial sequences had been available. The complete genome sequence we provide in this paper confirms it belongs to the Cheravirus genus, and allowed to unambiguously study relationships between members of the Cheravirus genus using full length polyprotein sequences. This allowed us to show that StPV is a different virus but forms together with AWPV and TgCV a distinct subgroup from other cheraviruses. Based on these information, we found interesting to include StPV in this paper as a Prunus-infecting cheravirus.

Reviewer 2 Report

The manuscript entitled "Molecular characterization of a new Prunus-infecting cheravirus and complete genome sequence of stocky prune virus" presents the sequence characterization of putative novel cheravirus discovered in Prunus species, as well as the complete sequence of the stocky prune virus. Although the manuscript presents originality and scientific significance, and most data is obtained from NGS technology, the use of simple phylogenetic analysis (neighbor-joining) in MEGA7 suite does not sounds robust enough to infer new virus species, as information regarding nucleotide substitution rates and evolutionary models are lacking, from my point of view. More robust and exaustive analyses are required to pinpoint new virus infecting Prunus species. Also, some incorrect english usage can be detected along the text, which require authors attention.

Author Response

The manuscript entitled "Molecular characterization of a new Prunus-infecting cheravirus and complete genome sequence of stocky prune virus" presents the sequence characterization of putative novel cheravirus discovered in Prunus species, as well as the complete sequence of the stocky prune virus. Although the manuscript presents originality and scientific significance, and most data is obtained from NGS technology, the use of simple phylogenetic analysis (neighbor-joining) in MEGA7 suite does not sounds robust enough to infer new virus species, as information regarding nucleotide substitution rates and evolutionary models are lacking, from my point of view. More robust and exaustive analyses are required to pinpoint new virus infecting Prunus species. Also, some incorrect english usage can be detected along the text, which require authors attention.

The molecular species demarcation criteria used in the family Secoviridae as defined in Thompson et al., (2017) and in the ICTV website (https://ictv.global/report/chapter/secoviridae/secoviridae) are the following: (i) CP amino acid sequence with less than 75% identity (for viruses with two or three CPs, combined CP sequences are considered) ; (ii) Conserved Pro-Pol region amino acid sequence with less than 80% identity ; (iii) For viruses with a bipartite genome, absence of re-assortment between RNA-1 and RNA-2. From our point of view, AWPV meet these criteria and should therefore be considered as a novel species.

Nevertheless, to take into account the reviewer’s comments, and in order to give some more robust elements to infer AWPV as a novel Prunus-infecting virus, we performed a phylogenetic reconstruction using the Maximum Likelihood method based on the aligment of amino acids of the two polyproteins (ClustalW), as ML is generally considered more reliable to infer phylogenetic relationships. The statistical significance of branches was evaluated by bootstrap analysis (1,000 replicates). The ML trees now replace the neighbor-joining trees in the revised manuscript (Figure 2A and 2B).

Reviewer 3 Report

None

Author Response

No comments from the reviewer 3

Reviewer 4 Report

In this study authors investigated viromes of several wild and cultivated Prunus species using HTS methods and identified a genetically divergent Cheravirus. Based on the genetic divergence of the virus authors proposed it as a tentative novel species called "alpine wild prunus virus (AWPV)" within the Cheravirus genus. Authors presented relevant data to show that the virus is phylogenetically distinct from other cheraviruses. The manuscript consists valuable information about a new virus in one of the important fruit tree crop, which adds value to the draft. However, the data requires better presentation. I feel the work can be presented better as a short report than a full-length article. 

L78, 80: double stranded "RNAs"....... I believe it could be used simply as RNA instead of RNAs

L37-38: "Members of " Split the  sentence

L240: It is not clear if authors verified the authenticity of the large 138 nt indel by RT PCR amplification. Even though the depth of coverage is very high, to eliminate the possibility of false contiging, I recommend authors to verify the indel by PCR amplification and sequencing.

Figure 1: spelling correction for "ttocky" and "Clivage"

L364: rephrase... "Given that few if any Prunus crops are grown where it was found would impose limits to its transfer to Prunus crops"

L365: replace "overall no doubt" with "no ambiguity"

Author Response

In this study authors investigated viromes of several wild and cultivated Prunus species using HTS methods and identified a genetically divergent Cheravirus. Based on the genetic divergence of the virus authors proposed it as a tentative novel species called "alpine wild prunus virus (AWPV)" within the Cheravirus genus. Authors presented relevant data to show that the virus is phylogenetically distinct from other cheraviruses. The manuscript consists valuable information about a new virus in one of the important fruit tree crop, which adds value to the draft. However, the data requires better presentation. I feel the work can be presented better as a short report than a full-length article. 

Our feeling is that a short report would be less appropriate to present the data included in the manuscript, with full genome sequence of two AWPV isolates and partial sequence for another one and in addition the first full genome sequence for stocky prune virus. As the other reviewers and the editor did not require such a format change and were apparently happy with a full paper format, we have selected to stay on a full-length article.

L78, 80: double stranded "RNAs"....... I believe it could be used simply as RNA instead of RNAs

This modification has been introduced across the whole manuscript.

L37-38: "Members of " Split the  sentence

Done

L240: It is not clear if authors verified the authenticity of the large 138 nt indel by RT PCR amplification. Even though the depth of coverage is very high, to eliminate the possibility of false contiging, I recommend authors to verify the indel by PCR amplification and sequencing.

A RT-PCR amplification was performed using total RNA of AWPV-Pb and AWPV-Pm as templates, and a primer pairs flanking the deletion. The PCR product was obtained in both cases and sequenced, confirming the deletion of 138 nt in AWPV-Pb RNA1.

Figure 1: spelling correction for "ttocky" and "Clivage"

Done

L364: rephrase... "Given that few if any Prunus crops are grown where it was found would impose limits to its transfer to Prunus crops"

We propose : « This virus was detected in an area with few or no Prunus crops, which may have prevented its transfer from wild to cultivated Prunus species. Ultimately, this could be one of the reasons why it was detected only once among more than 300 cultivated Prunus accessions tested by HTS during this survey. »

L365: replace "overall no doubt" with "no ambiguity"

Done

Round 2

Reviewer 4 Report

"A RT-PCR amplification was performed using total RNA of AWPV-Pb and AWPV-Pm as templates, and a primer pairs flanking the deletion. The PCR product was obtained in both cases and sequenced, confirming the deletion of 138 nt in AWPV-Pb RNA1."

Indicate the RT-PCR verification of the 138 nt indel in the main draft.

-If sticking to the full-length format, pay careful attention in presenting the data briefly and improving language used in the draft